# High Level of *Salmonella* Contamination of Leafy Vegetables Sold around the Niayes Zone of Senegal

Fatou Béye Sarré [1], Yakhya Dièye [2,3,*], Adji Marième Seck [1], Cheikh Fall [2] and Modou Dieng [4]

1 Laboratoire Nationale d'Analyses et de Contrôle des Produits Alimentaires, Rue Parchappe X Béranger Ferraud, BP 2050, Dakar 16000, Senegal
2 Pôle de Microbiologie, Institut Pasteur de Dakar, 36 Avenue Pasteur, BP 220, Dakar 12900, Senegal
3 Groupe de Recherche Biotechnologies Appliquées & Bioprocédés Environnementaux (GRBA-BE), École Supérieure Polytechnique, Université Cheikh Anta Diop, BP 5005, Dakar 10700, Senegal
4 Laboratoire d'Analyses et Essai, École Supérieure Polytechnique, Université Cheikh Anta Diop, BP 5005, Dakar 10700, Senegal
* Correspondence: yakhya.dieye@pasteur.sn; Tel.: +221-784578766

**Abstract:** Foodborne diseases are an important public health concern worldwide. Following a first global estimate of the burden of these diseases by the World Health Organization in 2015, many high-income countries implemented the generation of national data as a part of strategies to limit the transmission of pathogens through the food chain. In contrast, low-income countries, especially in the sub-Saharan Africa region, have limited data available on the burden and the source of contamination of produce destined for human and animal consumption. In this study, we analyzed *Salmonella* contamination of vegetables sold in supermarkets and open wet markets of five cities located in the Niayes zone, an area of high market gardening activities in the north western area of Senegal. We found high levels of *Salmonella* contamination of leafy vegetables, including mint leaves, parsley and lettuce. Contamination was higher in samples from supermarkets compared to open markets. Our results stress the need for an active surveillance of the microbiological quality of vegetables in Senegal, especially the presence of *Salmonella,* in order to prevent the risk of transmission of this bacterium through the food chain.

**Keywords:** foodborne diseases; vegetables; *Salmonella*; Senegal

## 1. Introduction

Foodborne diseases represent an important cause of mortality and morbidity worldwide [1]. In 2015, WHO reported, for the first time, estimates of the global burden of foodborne diseases showing 600 million cases of illness and 420,000 deaths for the year 2010 [2]. This study evaluated 31 hazards, including bacteria, viruses, parasites, toxins and chemicals, of which, microbes responsible of diarrheal and invasive infections were the most frequent cause of disease. Importantly, *Salmonella* was the first cause of death confirming data showing this bacterium as the most common foodborne pathogen worldwide and the need to implement measures aiming to detect, identify and prevent contamination along the food chain [2]. The WHO publication raised awareness about the need to generate quality data on the burden of foodborne pathogens. In high-income countries (HIC), there are systems that monitor contamination along the food chain, to which, the generation of data on the national burden was added [1]. In contrast, in low-and-middle-income (LMIC) countries, especially in the sub-Saharan Africa region, these data are largely unknown.

The species *Salmonella enterica* comprises over 2600 serotypes [3] widely present in various ecological niches. While some serotypes have a restricted host range, most of the *S. enterica* serotypes are capable of infecting a broad range of warm-blooded animals and humans [4]. *S. enterica* infects its host by the digestive route and primarily causes two types of disease: a gastroenteritis and an invasive infection. *Salmonella* gastroenteritis is

characterized by the development of the bacteria in the intestine and manifests as abdominal pain, diarrhea and vomiting [5]. It typically resolves after a few days in healthy adults. In contrast, it can lead to complications in children, the elderly and immunocompromised individuals [4]. Contrary to gastroenteritis, *Salmonella* invasive infection is a life-threatening disease that occurs when the bacteria invade the systemic compartment of the organism [6]. Invasive salmonellosis of humans include typhoid fever, caused by human-restricted serotypes Typhi and Paratyphi [7], and invasive nontyphoidal salmonellosis [8], caused by several broad host range serotypes. While *Salmonella* gastroenteritis is still a concern in both HIC and LMIC, invasive cases are very rare in the formers. In contrast, typhoid fever outbreaks and invasive nontyphoidal salmonellosis are frequent in low resource settings of sub-Saharan Africa and Asia, where poor sanitation and bad hygienic conditions prevail [8]. In this study, we wanted to assess the risk of contamination by *Salmonella* from vegetables in Senegal. We analyzed *Salmonella* presence in seven vegetables sold in supermarkets and open wet markets of five cities located in the Niayes area of Senegal, a zone of high market gardening activities.

## 2. Materials and Methods

### 2.1. Vegetables Acquisition and Processing

This is a prospective, random sampling study conducted in five cities located in or near the Niayes zone, a 180-km long coastal area located in north-western Senegal (Figure 1). The Niayes zone is an area of high market gardening activities, where 70% of Senegalese horticultural products are grown [9]. The sampling was conducted monthly between September 2020 and February 2021 in seven supermarkets and ten open wet markets, mostly supplied with products from the Niayes. The sampled products included cabbage, carrot, cucumber, lettuce, mint leaves, parsley and tomato. For supermarkets, the available vegetables were bought at the same sites during each sampling round. For open markets, the vegetables were purchased at different stalls, ensuring that only one sample of a given produce was purchased per vendor. The produces were sampled in sterile bags and immediately placed in a cooler until their shipment to the laboratory.

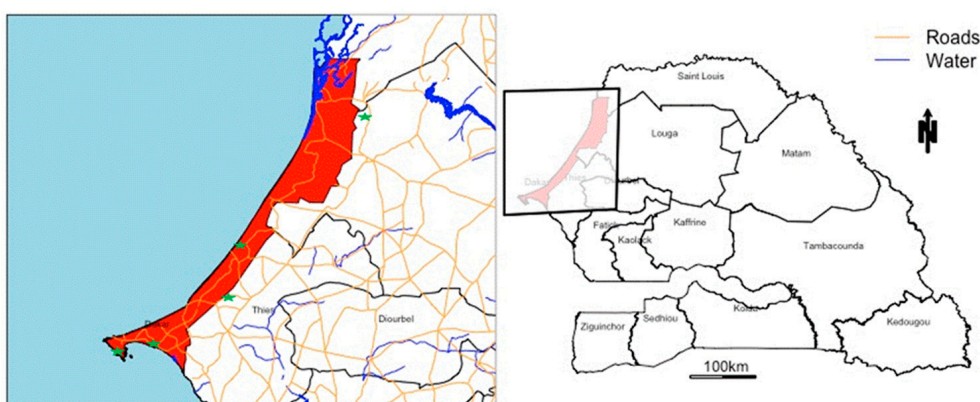

**Figure 1.** Localization of the Niayes zone (red) and cities (green stars) where vegetable sampling was conducted.

### 2.2. Isolation and Identification of Salmonella

Isolation and identification of *Salmonella* were performed according to ISO 6579-1 standard (Microbiology of the food chain—Horizontal method for the detection, enumeration and serotyping of *Salmonella*—Part 1: Detection of *Salmonella* spp.). Briefly, 25 g of vegetables were placed in 225 mL of buffered peptone water, homogenized using a Stomacher blender and incubated overnight at 37 °C. Following this, 1 mL and 0.1 mL of this pre-enriched sample were used to inoculate 10 mL of novobiocin-containing Muller Kaufman Tétrathionate and Rappoport Vassilidis broths, respectively. After 24 h of incubation at 37 °C, the broth contents were streaked onto Xylose Lysine Deoxycholate (XLD) and

Hektoen plates. Typical *Salmonella* colonies were confirmed by biochemical tests. Samples were considered free of *Salmonella* contamination when no colony of this bacterium was isolated from any of the inoculated plates.

### 2.3. Statistical Analysis

The statistical analyses were carried out using the IBM SPSS v28.0 software. Pearson Chi-Square was used to compare *Salmonella* prevalence among vegetables, sites and cities, with a *p* value < 0.05 being considered as statistically significant.

## 3. Results and Discussion

### 3.1. Study Design

To survey for *Salmonella* contamination of vegetables in Senegal, we conducted a prospective sampling in supermarkets and open markets in five cities located in or near the Niayes area of Senegal (Figure 1). The selected cities included Dakar (eight sampling sites), the capital city of Senegal, Rufisque (three sites), Thies (two sites), Mboro (one site) and Louga (three sites). The samplings were conducted monthly during a 6-month period between September 2020 and February 2021, with 110–144 samples collected each month. The sampled produces included cabbage (n = 103), carrot (n = 105), cucumber (n = 102), lettuce (n = 87), mint leaves (n = 79), parsley (n = 160) and tomato (n = 106) (Table 1).

**Table 1.** Contamination of vegetables by *Salmonella*.

| Mint | Parsley | Lettuce | Carrot | Cabbage | Cucumber | Tomatoes | Total |
|------|---------|---------|--------|---------|----------|----------|-------|
| 27/79 (34.2%) | 41/160 (25.6%) | 19/87 (21.8%) | 6/105 (5.7%) | 4/103 (3.9%) | 7/102 (6.9%) | 5/106 (4.7%) | 109/742 (14.7%) |

### 3.2. Differing Prevalence of Salmonella in Vegetables from Senegal

Of the 742 vegetable samples collected, 110 (14.8%) were contaminated by *Salmonella* (Table 1). Contamination was frequent in mint leaves (n = 27, 34.2%), parsley (n = 41, 25.6%) and lettuce (n = 19, 21.8%), while *Salmonella* was rarely present in the other produces analyzed (Table 1). Since mint leaves, parsley and lettuce are mostly consumed uncooked, these results reveal a high risk of human infection by *Salmonella* through these vegetables. *Salmonella* presence in leafy vegetables is not surprising since these products are known to be an important source of contamination by this bacterium [10]. Recent investigations in several sub-Saharan African countries including Burkina Faso [11], Ethiopia [12], Ghana [13] and South Africa [14] reported high prevalence of *Salmonella* in vegetables and fruits. Additionally, leafy vegetables including parsley and lettuce are particularly amenable to *Salmonella* colonization [15–17]. Indeed, laboratory investigations identified *Salmonella* genes involved in fimbriae biogenesis and biofilm formation as factors that promote colonization of plant organs and persistence [18–20]. The high level of contamination we found was not expected, though. A possible explanation for this result might be related to the use of wastewater or manure for vegetable production, which is a common practice in the Niayes. *Salmonella* contamination of vegetables grown using wastewater was reported in Burkina Faso [11], Ghana [21] and Morocco [22].

Next, we analyzed Salmonella presence in the most contaminated vegetables including mint, parsley and lettuce with regard to the cities and sampling sites (Table 2). *Salmonella* contamination of mint was significantly higher in Dakar (Person $\chi^2$ = 14.096, *p* = 0.007). In contrast, the level of contamination among cities was not significantly different for parsley and lettuce (Table 2). When considering the sampling sites, we noticed a higher level of contamination (>50%) of samples from supermarkets. We therefore compared *Salmonella* presence in samples from supermarkets and open markets (Table 3). Overall, samples from supermarkets were significantly more frequently contaminated than those from open markets (Person $\chi^2$ = 4.403, *p* = 0.036). This explains the higher level of contamination of samples from the city of Dakar that comprised 6/7 of the supermarkets included in

this study. When considering individual vegetables, significantly higher contamination was observed in mint samples only (Person $\chi^2$ = 10.874, *p* < 0.001) (Table 3). Two factors might contribute to the higher frequency of contamination of samples from supermarkets. Firstly, all the studied vegetables were mostly available in each supermarket, where they are manipulated by customers who inspect the products before making a choice. This practice can possibly favor both human transmission of bacteria and cross contamination of the vegetables. Secondly, vegetables from the supermarkets are conserved at relatively lower temperature and are presented to customers for a few days, while in open markets, the products are mostly sold the same day. Additional studies with more elaborated design are needed to address these questions.

**Table 2.** *Salmonella* contamination of mint, parsley and lettuce according to cities and sampling sites.

| City | Site | Mint | Parsley | Lettuce |
|---|---|---|---|---|
| Dakar | DK1 | 3/6 (50.0%) | 3/13 (23.1%) | 0/3 (0) |
| | DK2 | 2/5 (40.0%) | 7/17 (41.2%) | 1/4 (25.0%) |
| | DK3 | 2/2 (100%) | 3/12 (25.0%) | 2/6 (33.3%) |
| | DK4 | 5/6 (83.3%) | 6/11 (54.5%) | 2/6 (33.3%) |
| | DK5 | 2/5 (40.0%) | 2/13 (15.4%) | 1/5 (20.0%) |
| | DK6 | 2/3 (66.7%) | 0/8 (0) | 1/6 (16.7%) |
| | DK7 | 3/6 (50.0%) | 3/12 (25.0%) | 2/5 (40.0%) |
| | DK8 | 1/4 (25.0%) | 4/13 (30.8%) | 0/5 (0) |
| Total-DK | | 20/37 (54.1%) * | 28/99 (28.3%) | 9/40 (22.5%) |
| Rufisque | RF1 | 1/6 (16.7%) | 0/9 (0) | 2/4 (50.0%) |
| | RF2 | 1/1 (100%) | 3/9 (33.3%) | 1/6 (16.7%) |
| | RF3 | 1/8 (12.5%) | 2/13 (15.4%) | 0/6 (0) |
| Total-RF | | 3/15 (20.0%) * | 5/31 (16.1%) | 3/16 (18.8%) |
| Thies | TH1 | 0/6 (0) | 0/7 (0) | 2/6 (33.3%) |
| | TH2 | 1/3 (33.3%) | 2/7 (28.6%) | 0/6 (0) |
| Total-TH | | 1/9 (11.1%) * | 2/14 (14.3%) | 2/12 (16.7%) |
| Mboro | MB | 2/5 (40.0%) * | 0/3 (0.0%) | 2/5 (40.0%) |
| Louga | LG1 | 0/7 (0.0%) | 3/4 (75.0%) | 1/6 (16.7%) |
| | LG2 | 1/4 (25.0%) | 1/5 (20.0%) | 1/6 (16.7%) |
| | LG3 | 0/2 (0) | 2/4 (50.0%) | 1/2 (50.0%) |
| Total-LG | | 1/13 (7.7%) * | 6/13 (46.2%) | 3/14 (21.4%) |
| TOTAL | | 27/79 (34.2%) | 41/160 (25.6%) | 19/87 (21.8%) |

*, indicates statistically significant difference. Grey cases represent samples from supermarkets.

**Table 3.** Contamination of vegetables from supermarkets and open markets.

| *Salmonella* Presence | ALL | | Mint | | Parsley | | Lettuce | |
|---|---|---|---|---|---|---|---|---|
| | SM | OM | SM | OM | SM | OM | SM | OM |
| Yes | 54 | 55 | 17 | 10 | 23 | 18 | 7 | 12 |
| No | 246 | 387 | 13 | 39 | 58 | 61 | 29 | 39 |
| Percentage | 18% * | 12.4% * | 56.7% * | 20.4% * | 28.4% | 22.7 | 19.4% | 23.5% |

ALL, all vegetables; SM, supermarkets; OM, open markets. *, statistically significant difference (Pearson $\chi^2$).

## 4. Conclusions

Our study revealed a high risk of *Salmonella* contamination through leafy vegetables sold in supermarkets and open markets around the Niayes zone of Senegal. To our knowledge, this is the first report investigating *Salmonella* contamination of vegetables in Senegal. Our results stress the need for additional investigations in order to identify the origins of contaminating bacteria and to establish strategies to prevent the acquisition of *Salmonella* through consumption of raw or uncooked vegetables. Besides the microbiological approach used in this work, additional studies can include molecular techniques for the rapid detec-

tion of *Salmonella* [23]. The Niayes is a zone of high market gardening, with its produces used to supply several cities of Western Senegal and beyond, and with the potential to export horticultural products outside of Senegal. For this reason, policymakers need to establish and enforce rules that limit the risks of contamination by foodborne pathogens.

**Author Contributions:** Conceptualization, F.B.S. and A.M.S.; formal analysis, Y.D. and F.B.S.; investigation, F.B.S. and A.M.S.; resources, C.F. and M.D.; writing—original draft preparation, Y.D.; writing—C.F. and M.D. All authors have read and agreed to the published version of the manuscript.

**Funding:** This research was supported by the Senegalese Ministry of Health and the Institut Pasteur de Dakar, Senegal.

**Data Availability Statement:** All the data reported in this study are available at the Laboratoire Nationale d'Analyses et de Contrôle des produits alimentaires, Ministry of Trade and Small and Medium Companies, Senegal.

**Conflicts of Interest:** The authors declare no conflict of interest.

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
