# Peer review of "High Level of Salmonella Contamination of Leafy Vegetables Sold around the Niayes Zone of Senegal"

_horticulturae, doi:10.3390/horticulturae9010097_

Round 1

Reviewer 1 Report

The expression is clear. However, some minor errors need to be corrected. A major concern is that, as authors mentioned, Salmonella enterica comprises thousands of serotypes, some harm to people, some not. Which are the serotypes of Salmonella enterica isolated by authors, whether the isolated Salmonella enterica are harm to human? This is the most important and fundamental information for the study, and this is also the most meaningful information for readers. Other miner concerns are following.

L33. Reference was cited in [], not in (). Thoroughly check the manuscript.

L86. Should be ‘mL’. Thoroughly check the manuscript.

L97. ‘.05’ or ‘0.05’? The ‘p’ should be italic. Thoroughly check the manuscript, such as L126, L132.

Reviewer 2 Report

The paper is timely and scientifically sound. Indeed, Salmonella sp. is one of the most widely distributed cause of dangerous foodborne diseases and must be continuously monitored in different regions of the world, including Africa. Therefore, this work is important for experts in the areas of food safety, microbiology and quarantine. At the same time, I have to note that more informantion regarding different approaches (other than microbiological and biochemical tests) of Salmonella detection as well as additional data (described in recent publications) on levels of Salmonella contamination in different regions of Africa would make this manuscript much more interesting and significant.

For example, there are some papers devoted to detection and identification of Salmonella sp. based on different types of PCR, e.g. in African countries (Tennant et al., 2010; Al-Emran et al., 2016; Mthembu et al., 2019; Kagambèga et al., 2021 etc.). In the ideal case, the authors could use some of these molecular detection systems to confirm the results of biochemical and microbiological identification and characterisation of pathogens isolated from vegetables analyzed in this study. At least, those approaches should be noted in Introduction and/or in a chapter where results are discussed. 

Also, it would be great if the authors give more information about previous studies of Salmonella levels in Africa. For example, a chapter "Microbial and chemical contamination of vegetables in urban and peri-urban areas of Sub-Sahara Africa" by Traore et al. (in book: Climate change - recent observations) can be mentioned and included in the Reference list. 

Reviewer 3 Report

I had the opportunity to review the Submission horticulturae-2127006. The paper is about evaluate the Salmonella contamination of leafy vegetables sold around the Niayes zone of Senegal. In this study, author analyzed Salmonella contamination of vegetables sold in supermarkets and open wet markets of five cities located in the Niayes zone, an area of high market gardening activities in North Western area of Senegal. The results stress the need of an active surveillance of the microbiological quality of vegetables in Senegal, especially the presence of Salmonella in order to prevent the risk of transmission of this bacterium through the food chain. The study is of interest but needs to be completed. I consider that it presents limitations, so my recommendation is minor revision.

Main aspects that support my recommendation

i) Salmonella infection (salmonellosis) is a common bacterial disease that affects the intestinal tract. Salmonella bacteria typically live in animal and human intestines and are shed through stool (feces). Humans become infected most frequently through contaminated water or food. Salmonella infection is usually caused by eating raw or undercooked meat, poultry, and eggs or egg products or by drinking unpasteurized milk. Salmonella can be killed quickly in five or six minutes at 70℃. The higher the temperature, the better the sterilization effect. The United States Department of Agriculture requires chilled poultry carcass temperature to be below 4°C (40°F) to inhibit the growth of Salmonella and improve shelf life. Salmonella can grow aerobically or anaerobically depending upon conditions, 37°C (98.6°F) and a pH of 6.5 to 7.5 are optimal. Did the authors consider the temperature of sample transportation and storage when selecting the samples? I think the temperature is very important for the results of the later test, especially the storage temperature.

ii) There are many types of Salmonella, and the pathogenicity of different genera is not the same. I think strain pathogenicity should also be considered in this study.

iii) I strongly recommend that the authors perform 16S amplicon analysis on the samples. In this way, both the species and the abundance of Salmonella in samples from different places can be known.

iv) Why did the authors not identify the colonies by PCR? Because sometimes there are false positives.

Round 2

Reviewer 1 Report

Nice work.

Reviewer 2 Report

The reviewed version of the MS contains the references I suggested to include and additional information needed to describe the 'state-of-art' of Salmonella contamination in different regions of Africa as well as different approaches of detection of this pathogen. So I believe the MS can be accepted in present form.